



# Key ingredients in regional climate modeling for improving the representation of typhoon tracks and intensities

Qi Sun[1], Patrick Olschewski[1], Jianhui Wei[1], Zhan Tian[3,4], Laixiang Sun[5,6], Harald Kunstmann[1,2], Patrick Laux[1,2]

[1]Institute of Meteorology and Climate Research (IMK-IFU), Karlsruhe Institute of Technology, Campus Alpin, Garmisch-Partenkirchen, Germany
[2]Institute of Geography, University of Augsburg, Augsburg, Germany
[3]School of Environmental Science and Engineering, Southern University of Science and Technology, Shenzhen, China
[4]Pengcheng Laboratory, Shenzhen, China
[5]Department of Geographical Sciences, University of Maryland, College Park, MD, USA
[6]School of Finance & Management, SOAS University of London, London, UK

*Correspondence to*: Qi Sun (qi.sun@kit.edu)

**Abstract.** There is evidence of an increased frequency of rapid intensification events of tropical cyclones (TCs) in global offshore regions. This will not only result in increased peak wind speeds but may lead to more intense heavy precipitation

events, leading to flooding in coastal regions. Therefore, high impacts are expected for urban agglomerations in coastal regions such as the densely-populated Pearl River Delta (PRD) in China. Regional climate models (RCMs) such as the Weather Research and Forecasting (WRF) model are state-of-the-art tools commonly applied to predict TCs. However, typhoon simulations are connected with high uncertainties due to the high number of parameterization schemes of relevant physical processes (including possible interactions between the parameterization schemes) such as Cumulus (CU) and Micro Physics

(MP), and other crucial model settings such as domain setup, initial times, and spectral nudging. Since previous studies mostly focus on either individual typhoon cases or individual parameterization schemes, in this study a more comprehensive analysis is provided by considering four different typhoons of different intensity categories with landfall near the PRD, i.e., Neoguri (2008), Hagupit (2008), Hato (2017), and Usagi (2013), as well as two different schemes for Cu and MP, respectively. Moreover, the impact of the model initialization and the driving data is studied by using three different initial times and two

spectral nudging settings. Compared with the best-track reference data, the results show that four typhoons show some consistency. For track bias, nudging only horizontal wind has a positive effect on reducing the track distance error; for intensity, compared with a convective-permitting (CP; nudging potential temperature and horizontal wind; late initial time) model, using Kain-Fritsch scheme (KF; nudging only horizontal wind; early initial time) configuration shows relatively lower minimum sea level pressures and higher maximum wind speeds which means stronger typhoon intensity. Intensity shows less sensitivity to

two MP schemes compared with the CP, nudging, and initial time settings. Furthermore, we found that compared with the CP, using the KF scheme shows a relatively larger latent heat flux and higher equivalent potential temperature, providing more energy to typhoon development and inducing stronger TC. This study could be used as a reference to configure WRF for historical and future TC simulations and also contributes to a better understanding of the model simulation performance of principal TC structures.


Keywords: Tropical cyclone; WRF configurations. Model output optimization; Pearl River Delta region.

## 1 Introduction

Climate change influences extreme weather and climate events. Tropical cyclones (TC) are one extreme event that shows a

significant response to global warming. Over the past four decades, there is evidence of a globally increased frequency of TC rapid intensification events in global offshore regions (Li et al., 2023). According to the IPCC AR6 report, the impact of global





warming can be observed in the average and maximum precipitation rates associated with TCs, showing an upward trend across the globe (Seneviratne et al.,2021). Moreover, the occurrence of intense TCs rises with global warming, as well as an increase in average peak TC wind speeds and the maximum wind speeds (MWS) of the most severe TCs on a global scale.

This may pose significant social threats to the regions affected by the TCs (Knutson et al., 2020; Murakami et al., 2017). Guangdong Province is located in the southeastern coastal region of China and is the area where TC activity is most frequent, with the greatest impact and the longest duration throughout the year (Tang et al., 2014). The Pearl River Delta (PRD) region, which is located in the south of Guangdong, is one of the most prosperous economic areas with its gross domestic product (GDP) of more than $1 trillion and a population exceeding 70 million individuals in 2021 (Statistical Bureau of Guangdong

Province, 2023). According to historical data and meteorological observations, the PRD region is typically affected by an average of 3-5 typhoons per year (Hong Kong Observatory, 2023). TCs often have substantial societal impacts, for example, the strong winds associated with TCs can cause structural damage to buildings, especially in regions in which construction land is highly concentrated. The heavy rainfall brought by TCs can lead to flashfloods and storm surges associated with these cyclones can cause coastal erosion. For example, when Typhoon Hato made landfall, it caused a maximum storm surge of 2.79

m in Zhuhai, causing significant damage to areas such as Zhuhai, Hong Kong, and Macau and resulting in 24 deaths and an economic loss of 6.82 billion US dollars (Hong Kong Observatory, 2017). Above all, accurate prediction of typhoons is of great social significance for the development of the region.

Regional climate models (RCMs) such as the Weather Research and Forecasting model (WRF) are commonly used to forecast Tropical cyclones (TC). This is possible because TCs are mesoscale atmospheric systems and high-resolution models are able

to represent convection and the other physical processes of a TC system relatively well (Moon et al., 2018). For example, Gutmann et al. (2018) reproduced 30 out of 32 named storms using WRF and simulated the TC tracks, storm radii, and translation speeds well, despite the MWS being simulated lower than observed. Other researchers pointed out the accurate prediction of TC characteristics because WRF can reasonably capture atmospheric circulation patterns such as subtropical high, steering flows, and vertical wind shear (Xu et al., 2023), which is highly related to the motion and structure of Typhoon

Lekima. Besides, the heat energy exchange between ocean and air, especially latent heat flux, are important energy source for TCs which is also highly related to TC intensity. During the TCs intensification stages, the latent heat flux from the ocean to the atmosphere increases (Chen et al., 2014), and TCs absorb latent heat, increasing the available potential energy. Later, a part of the latent energy is released in convective clouds, increasing the kinetic energy (Ma et al., 2015). Sun et al. (2019) concluded that WRF produced relatively good performances in intensity as it is able to roughly resolve the ocean–TC

interactions through latent and sensible heat energy exchange. Above all, the simulation of typhoons using high-resolution regional climate models can capture TC characteristics and has already been widely applied in various fields. Track prediction could be used for regional exposure extent and disaster early warning, rainfall, and wind prediction could also be used for city disaster management and defense. Furthermore, it is also an important tool for dynamically downscaling climate models with around 100–200 km grid spacing, e.g. participants of the Coupled Model Intercomparison Project, Phase 6 (CMIP6) multi-

model projections, to project future TC activities e.g., genesis, frequency, intensity, tracks, precipitation, and future TC-induced flood risks analysis. Compared with WRF, these coupled models are too coarse and are limited in capturing the detailed atmospheric circulation patterns and ocean-TC interactions, which affect TC tracks and intensity.

However, previous research (e.g., Sun et al., 2015) mentioned that WRF itself shows many uncertainties within simulating TCs because of different configurations in the horizontal grid spacing (Gentry and Lackmann, 2010; Sun et a.,2015), a

combination of physical schemes such as cumulus (CU), microphysics (MP) and planetary boundary layer (PBL) parameterization schemes (Sun et al., 2019; Delfino et al., 2022; Bhattacharya et al., 2017; Khain et al., 2016; Shepherd and Walsh, 2017; Zhang and Wang, 2018; Zhang et al., 2022), initial and boundary conditions (Raktham et al., 2015; Xu et al., 2023), initial times (Delfino et al., 2022), and spectral nudging settings (Mori et al., 2014; Moon et al., 2018), substrate conditions (Zhang et al., 2023, 2019) after landfall. For example, as for horizontal grid spacing, Gentry and Lackmann (2010)





conducted sensitivity simulations of Hurricane Ivan (2004) using horizontal resolutions between 12-2 km and the results demonstrated that the model solution for the structure and intensity exhibits partial convergence at grid spacings ranging from 8-4 km, indicating that these spacings could be suitable for operational numerical weather prediction (NWP) applications, and intensity differences of only around 10 hPa between them. Sun et al. (2013) used WRF to simulate Typhoon Shanshan (2006) with changes in horizontal grid spacing at gray-zone resolutions (7.5–1 km) and the results revealed that the intensity of the

TC shows a relatively small change as the grid spacing decreases from 5-3 km, while a significant increase was found from 3-1 km. However, the fine resolution has a larger bias in intensity compared with the coarser resolution. The former two examples used nested domains to conduct simulations, however, Gutmann et al. (2018) directly used a single large 4 km domain to simulate 32 TCs and concluded that the model can realistically reproduce most of the major TCs. One single convection-permitting domain with a resolution of 4.5 km shows no significant difference compared to the results of Delfino et al. (2022),

who simulated Typhoon Haiyan using an inner domain 5 km simulation, nested in a 25 km outer domain. Above all, the horizontal resolution shows a great impact on TC intensity. Former studies already did extensive research on the analysis of TC sensitivities. However, previous studies on typhoon parameterization sensitivity mainly focused on individual typhoon cases, which may lack representativeness. Although some researchers have selected multiple typhoons for their studies, their research primarily focuses on conducting experimental studies on individual parameterizations because of computational costs.

Therefore, it is of great importance to understand the sensitivity of numerous model configurations for different intensities of TCs before application in future impacts.

Convective processes, such as CU convection, play a crucial role in the life span of TCs and serve as the primary source of energy for their occurrence and development (Camargo and Wing, 2016). Convective processes influence sensible and latent heat and momentum transport and then affect the vertical structures of atmospheric temperature and humidity fields (Anthes,

1977; Li et al., 2018; Zhang et al., 2021) which greatly influences TC intensity and track (Sun et al., 2019). Because the resolution of the model is relatively coarse and cannot explicitly represent the convection process, some researchers developed different CU parameterization schemes to represent these processes. Prior research usually compares different CU schemes to investigate the influence of parameterizations on the track and intensity of simulated TCs (Delfino et al., 2022; Sun et al., 2019; Sun et al., 2015; Li et al., 2018). Most of the results show that, compared with other schemes, using the Kain-Fritsch

(KF; Kain, 2004) scheme is in best agreement with the observation because it reasonably represents shallow and deep convection above the ocean surface. Other researchers applied WRF specifically permitting convection using horizontal resolutions below 5 km (CP; Gutmann et al., 2018; Gentry and Lackmann, 2010). However, modeling physical processes in the gray zone of horizontal resolutions ranging from 1 km-10 km grid-spacing, despite exhibiting explicit convective processes, may be insufficient to faithfully capture the entire range of convective motions (Bryan et al., 2003). By simulating TC Haiyan,

with a 4 km horizontal resolution using the KF scheme and for 2 km using no CU parameterization scheme better reproducing TC (Li et al., 2018). Above all, in this study, we conducted sensitivity tests using the KF scheme, as well as applying WRF as a convection-permitting model, to simulate TCs of different intensities using a single 5 km domain to find a configuration that can realistically reproduce TC intensity and structure.

Previous research (e.g., Sun et al.,2015; Sun et al., 2019) illustrated that MP schemes also show an impact on TC simulations

which may be induced by different explicitly resolved moisture species and physical processes involved in the phase changes (Thompson et al., 2008). Based on the previous research, two schemes: WRF Single-Moment 6-class (WSM6; Hong et al., 2004), and New Thompson (Thompson et al., 2008) are commonly used in WRF simulation. These two schemes have the same number of mass variables, but Thompson also takes the number of concentrations for rain and ice species into consideration which may impact the rainfall process in the TC system, thus impacting the latent heat release and therefore the

intensity. As mentioned previously by Sun et al. (2019), these two schemes show a difference in simulated MWS and minimum sea level pressure (MSLP) for Typhoon Hagupit because of the differences in latent and sensible heat flux between ocean and air. In this study, we tested TCs of different intensities for different combinations of CU and MP parameterization schemes.





As for the initial times, researchers use different initial times, e.g., 6-12 h for short-time simulations, and still lack consensus on the spin-up time (Liu et al., 2023), imposing large uncertainty on simulations, especially for extreme events. Mooney et al.

(2019) summarized accurately represent strong TCs intensity at the initial time is important in subsequent TC simulation. In this study, we selected the initial time according to different TC intensities. Based on the definition of TC intensity, the three stages before a TC reaches typhoon intensity are chosen based on six-hour observation records of TCs (e.g., CMA; Ying et al., 2014): the last time of tropical depression (TD), the beginning time of tropical storm (TS), and the beginning time of severe tropical storm (ST) to determine the optimal initial time for typhoon simulation. Different TCs have different lengths of

intensification periods before attaining typhoon intensity. Therefore, simulating different TC cases may also show if model performance is only related to the duration of spin-up time or also related to the initial TC intensity.

Spectral nudging is a technique that consists of driving RCMs on selected spatial scales corresponding to those produced by the driving fields and prevents large and unrealistic departures between the driving fields and the RCM fields at the driving fields' spatial scales (Omrani et al., 2012). This technique is commonly used for WRF simulations and plays a crucial role in

enhancing the performance of dynamical downscaling in TC simulations (Mori et al., 2014; Delfino et al., 2022; Chen et al., 2020; Moon et al., 2018; Cha et al., 2011; Kueh et al., 2019). Extensive research concluded that the spectral nudging technique exhibits the ability to improve track bias by influencing large-scale steering flow which has a great impact on TC tracks. For example, Delfino et al. (2020) pointed out that the nudging technique can improve the mean track bias of Typhoon Haiyan by 20 km. However, compared with the track improvement, the nudging technique shows detrimental effects on TC intensities.

For example, Cha et al. (2011) demonstrated that spectral nudging leads to a reduction in the intensities of simulated typhoons by inhibiting the development process of typhoons. TC intensities are also influenced by small-scale processes. These intrinsic small-scale processes are reproduced by the WRF model and the nudging technique impedes their development process because this information does not exist in the large-scale driving field. Former researchers mainly focused on broadening nudging intervals (Cha et al., 2011), using small weighting, and different cutoff wavelengths (Moon et al., 2018; Mai et al.,

2020; Gómez and Miguez-Macho, 2017) to improve performance. However, few publications focus on the sensitivity of nudged components. Different researchers nudged different variables which may also cause uncertainties in the results. For example, Delfino et al. (2020) nudged the horizontal and vertical wind components, the potential temperature, and the geopotential height above the PBL. Kueh et al. (2019) applied the nudging technique to the horizontal wind components, potential temperature, and water vapor mixing ratio above the PBL. Moon et al. (2018) pointed out that the effect of humidity

is not as significant as in other fields. Chen et al. (2020) only nudged the model horizontal wind above 500 hPa to provide a realistic steering flow and to prevent an influence on the inner core circulations of the simulated TCs. Although differences exist regarding included variables, all the publications consistently use this technique above the PBL. Furthermore, the nudging effect is dependent on the region of the TC track, and the technique was especially effective for TCs that occurred to the east of the Western North Pacific (WNP) and turned near the Northeast Asia (Moon et al., 2018). However, for most of the TC

tracks located outside of the South China Sea (SCS), the effect of the technique on TCs formed in the SCS region is not clear, which accounts for approximately 30% of the total number of TCs affecting China (Cao et al., 2020). Above all, in this study, we conducted sensitivity tests for nudging different variables above 500 hPa and its impact on TC intensity and track, while also considering TC genesis.

The main objective of this study is to analyze how well WRF is able to represent TCs affecting the PRD region with different

intensities and genesis locations. More specifically, it analyzes (i) how sensitive are the typhoon simulations to CU and MP parameterization schemes, initialization time, and spectral nudging variables, and (ii) how sensitive the above-mentioned settings are in reproducing typhoons belonging to specific intensity categories or genesis locations.





## 2 Methods and data

This section consists of four subsections: first the models used in the study and the experimental design is introduced, followed
by an overview of Typhoon Cases and the introduction of the data used for validation and the tracking algorithm for TC
detection.

### 2.1 Model description and experimental design

In this study, the Advanced Research Weather Research and Forecasting (WRF-ARW) model version 4.3.3 (Skamarock et al.,
2019) was used to conduct typhoon simulations. Based on previous sensitivity research on horizontal resolution (Gutmann et
al., 2018; Sun et al., 2013; Gentry and Lackmann, 2010), the model was set up with one single 5 km domain to better reproduce
TC intensity and structure. The center point was located at 18.5°N, 124.0°E, with a regional grid of 550×950 covering all
typhoon tracks (Figure 1). The model top was at 50 hPa, and 52 sigma layers were used in the vertical. For shortwave radiation
and longwave radiation, in this study, we used the Dudhia scheme (Dudhia, 1989) and the Rapid Radiative Transfer Model
scheme (RRTM) (Mlawer et al., 1997). For the planetary boundary layer scheme, the Yonsei University nonlocal PBL scheme
(Hong et al., 2006) with a surface boundary layer scheme based on Zhang and Anthes (1982) was used. For the land surface
scheme, we used the unified Noah Land Surface Model (Chen and Dudhia, 2001). The initial and boundary conditions were
interpolated from the European Centre for Medium-Range Weather Forecasts (ECMWF) Reanalysis version 5 (ERA5) with
0.25°× 0.25° spatial resolution and 6-hour temporal resolution (Hersbach et al., 2020). The land surface information was
obtained from the Moderate Resolution Imaging Spectroradiometer (MODIS) satellite dataset with 20 land use classifications.
In this study, all applied configurations are given in Table 2. Since the 5 km spatial resolution is within the grey zone for the
CU scheme, we compare the impact of the KF scheme and a CP model for the different TCs. In terms of the MP scheme,
WSM6 and Thompson are tested in this study. For spectral nudging, we investigate nudging only horizontal wind above 500
hPa as well as nudging horizontal wind and potential temperature. For the initial time, we use the TC intensity definitions to
start the simulation in an attempt to assess which initial time will produce the most accurate results.

**Table 1 The abbreviations of configurations used in the sensitivity experiments.**

| Cumulus schemes | Microphysics schemes | Nudging variables | The initial time of simulation | | |
| --- | --- | --- | --- | --- | --- |
| | | | TD | TS | ST |
| KF(KF) | Thompson (TH) | Potential temperature, U and V wind (PT+UV) | KF-TH-(PT+UV)-TD | KF-TH-(PT+UV)-TS | KF-TH-(PT+UV)-ST |
| | | U and V wind (UV) | KF-TH-(UV)-TD | KF-TH-(UV)-TS | KF-TH-(UV)-ST |
| | WMS6 (W6) | Potential temperature, U and V wind (PT+UV) | KF-W6-(PT+UV)-TD | KF-W6-(PT+UV)-TS | KF-W6-(PT+UV)-ST |
| | | U and V wind (UV) | KF-W6-(UV)-TD | KF-W6-(UV)-TS | KF-W6-(UV)-ST |
| Convection-permitting (CP) | Thompson (TH) | Potential temperature, U and V wind (PT+UV) | CP-TH-(PT+UV)-TD | CP-TH-(PT+UV)-TS | CP-TH-(PT+UV)-ST |
| | | U and V wind (UV) | CP-TH-(UV)-TD | CP-TH-(UV)-TS | CP-TH-(UV)-ST |
| | WMS6 (W6) | Potential temperature, U and V wind (PT+UV) | CP-W6-(PT+UV)-TD | CP-W6-(PT+UV)-TS | CP-W6-(PT+UV)-ST |
| | | U and V wind (UV) | CP-W6-(UV)-TD | CP-W6-(UV)-TS | CP-W6-(UV)-ST |



### 2.2 Overview of Typhoon cases

For this study, we chose different intensities of TCs based on the Saffir-Simpson scale (Simpson and Saffir, 1974) which affect the PRD region, and induced compound flood events (e.g., TC-induced heavy rainfall, strong wind-induced storm surge,
accompanied by the occurrence of astronomical tide). We also considered the origin of TC genesis, one TC originated formed in SCS region, and the latter three TCs were formed in the Northwest Pacific Ocean. Based on this, we chose Typhoon Neoguri (2008), Typhoon Hagupit (2008), Typhoon Hato (2017), and Typhoon Usagi (2013). Figure 1 and Table 2 briefly describe the four TC cases.

**Table 2 Different intensity typhoons affecting the PRD region.**

| TC No. | TC name | Category MWS (m s$^{-1}$) | Initialization Time (yyyy-mm-dd-hh)/MSLP (hPa)/ MWS (m s$^{-1}$) | | | End of simulation |
|---|---|---|---|---|---|---|
| | | | Last TD | TS | ST | |
| 1 | Neoguri | 1(40) | 2008-04-15-00 UTC 1002/15 | 2008-04-15-06 UTC 998/18 | 2008-04-16-00 UTC 990/25 | 2008-04-20-00 |
| 2 | Hagupit | 2(50) | 2008-09-19-06 UTC 1004/15 | 2008-09-19-12 UTC 1002/18 | 2008-09-20-06 UTC 990/25 | 2008-09-25-12 |
| 3 | Hato | 3(52) | 2017-08-20-00 UTC 1002/15 | 2017-08-20-06 UTC 998/18 | 2017-08-22-00 UTC 985/25 | 2017-08-24-12 |
| 4 | Usagi | 4(60) | 2013-09-16-12 UTC 1002/15 | 2013-09-16-18 UTC 1000/18 | 2013-09-18-00 UTC 985/25 | 2013-09-23-06 |


As shown in Figure 1 and Table 2, Typhoon Neoguri originated off the west coast of Mindanao Island. Propagating westward and passing over the Sulu Sea and then moving gradually to the northwest. It reached typhoon intensity over the middle of the SCS region six hours later and advanced towards the north. After traversing the east of Hainan Island, its intensity gradually diminished. The difference between Typhoon Neoguri and the other considered storms is, that the latter three TCs formed in
the Northwest Pacific Ocean where they gained more energy and attained relatively higher intensity.

Typhoon Hagupit originated at the west of the Mariana Islands and then proceeded to move in a west-southwest direction. Later, it intensified into a typhoon moving towards the northwest. Gradually, the typhoon proceeded in a west-northwest direction and crossed the northern coastline of Luzon Island. After landfall, Hagupit weakened and then continued to move west-northwest.

Typhoon Hato originated over the sea east of the Philippines and moved westward. After passing the Luzon Strait, it attained typhoon intensity over the northern part of the South China Sea and further intensified into a STY. Three hours later it attained Super Typhoon (SuperTY) intensity. Typhoon Hato made landfall along the southern coast of Zhuhai in Guangdong Province. After hitting the southern part of China, Hato rapidly weakened to TD intensity.

Typhoon Usagi formed as a TD southwest of Okinot Orishima Island and slowly moved eastward. After turning westward
over the same ocean, it strengthened into TS intensity. Moving slowly westward, Usagi was upgraded to typhoon intensity east of the Philippines. It developed rapidly and reached STY intensity and SuperTY intensity another 6 hours later. Keeping its west-northwestward track, Usagi passed through the Luzon Strait and entered the SCS. It made landfall in the southern part of China with TY intensity the next day and was weakened to TD intensity.



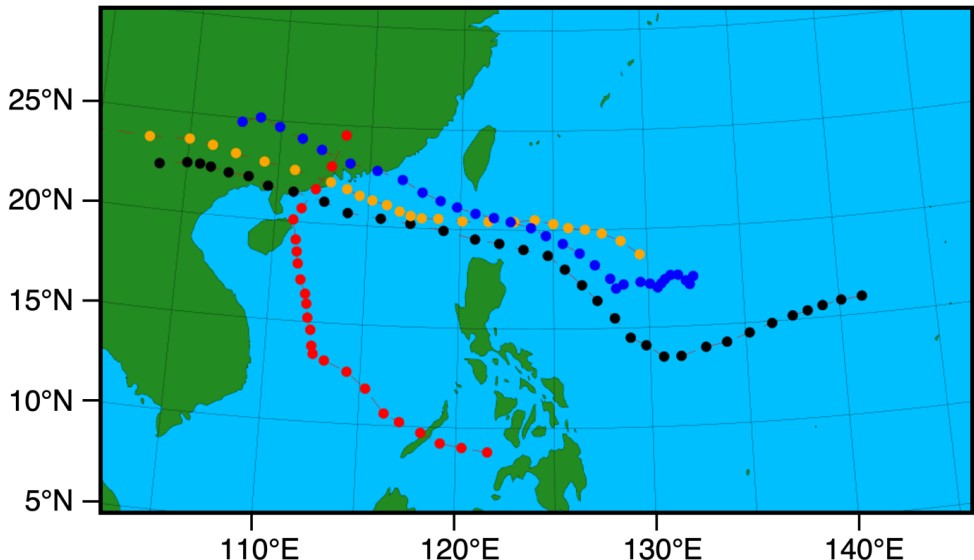

**Figure 1: The domain used in this study with a single 5 km horizontal resolution. The red dashed line represents Neoguri's full track from TD formation based on CMA data, the black line represents Hagupit, the orange line represents Hato, and the blue line represents Usagi. The PRD region is located at around 21°-24°N, 111.5°-115.5°E.**

## 2.2 Validation data

The previous research summarized that in the WNP basin, the best-track datasets are inconsistent in various aspects, including wind averaging times, storm position, and intensity (Ying et al., 2014). Here, we use three reference data sets from different agencies including the China Meteorological Administration (CMA; Ying et al., 2014; Lu et al., 2021), the World Meteorological Organization (WMO) Regional Specialized Meteorological Center in Tokyo, Japan (JMA,2023; https://www.jma.go.jp/jma/jma-eng/jma-center/rsmc-hp-pub-eg/Besttracks/e_format_bst.html), and the Oceanographic Data Center, Chinese Academy of Sciences (CAS; Wang., 2013), to validate the model performance. All data sets include the time, category, longitude, latitude, MSW, and MSLP information of the TCs.

To evaluate the performance of simulated rainfall in terms of temporal and spatial patterns, we used GPM IMERG Final Precipitation L3 V06 data with a half-hourly time resolution and 0.1° x 0.1 °spatial resolution (Huffman et al.,2019). This data is commonly used for the validation of TC rainfall simulations.

## 2.4 Tracking algorithm

The tracking algorithm used in this study is adopted from Gutmann et al.(2018). However, in this study, the 400 km x 400 km evaluation box around the storm center is substituted by a 4° x 4° box on a regular lon-lat grid. The MSLP and MWS are diagnosed when reached the thresholds that SLP is 27 hPa below long-term SLP and wind speed is more than 25 m s$^{-1}$ in WRF and subsequently used for the detection of TC information. The MWS and precipitation rates are the local maximum values around the storm center in the 4° x 4° evaluation region. The MSLP detected by the algorithm is the minimum pressure in the region.

## 3 Result and Discussion

The results are presented as tracks of the four typhoons, MSLP, MWS, rainfall, latent heat flux, and equivalent potential temperature sensitivity for different configurations in WRF.





### 3.1 TC tracks

Figure 2 shows that the typhoon tracks simulated by the 24 experiments for the four typhoons of different intensities and genesis were consistent with the track information from the CAS, CMA, and JMA best tracks. The best tracks of the four typhoons plotted in Figure 2 start around the formation of the TD showing the different genesis locations of the four typhoons. Within the TC formation phase, some deviations can be detected for the three best tracks, however, after attaining typhoon intensity, they are mostly consistent in terms of typhoon tracks. The simulation results plotted in Figure 2 start when the

prerequisites of the typhoon tracking algorithm are met.

From Figure 2a, no large differences between the 24 experiments can be seen, and the simulated tracks inherit a large consistency with the best tracks. Because of the influence of the nudged environmental steering flow, typhoons gradually move northward after formation. However, around 20° N, the three best tracks show a slight shift to the west which could not be captured by all the model simulation results. In addition, after landfall, the simulated tracks vary remarkably which may be

due to the sensitivity of the 24 settings to the land surface.

Nudging horizontal wind could greatly improve typhoon tracks by impacting on large-scale circulation patterns and steering flow. This finding is in accordance with previous studies (e.g., Mori et al., 2014; Delfino et al., 2022; Chen et al., 2020; Moon et al., 2018; Cha et al., 2011; Kueh et al., 2019). To show the nudging effects on tracks in the SCS region, we conducted an additional sensitivity test without nudging (W6-KF-00-TD) for the four TCs. Figure S1a shows that for Typhoon Neoguri,

without nudging, the simulated track shifts to the west, passing through the middle of Hainan Island with a track bias of more than 2°. Thus, the nudging technique also shows improvement in the SCS region.

Figures 2b-d show that the tracks of Typhoon Hagupit, Hato, and Usagi inherit a certain degree of similarity. The genesis of the TCs was east of the Philippine islands and subsequently, the TCs gradually moved towards the west and crossed the ocean between Taiwan and the Philippine islands. Figures 2b-d show that there are only slight differences between the 24 experiments.

Also, the simulated tracks are in good agreement with the best tracks. For Typhoon Hagupit, in Figure S1b, the no-nudging track is shifted to the north, past Taiwan Island, and the landfall location is east of the PRD region, compared with the best track. As for Typhoon Hato, in Figure S1c the no-nudging track is shifted to the northern region and the bias regarding the best track is around 1-2° which also shows that the nudging technique could improve the track accuracy. For Typhoon Usagi, in Figure S1d the no-nudging track is shifted to the north in the early stages, then shifted to the south and close to the best tracks,

subsequently however shifted back north and finally depicts landfall far away from the PRD region.

Compared with CMA, the average bias is around 0-0.6°, except for two experiments of Hagupit (TH_CP_(UV)_ST, W6_CP_(PT+UV)_ST) inheriting a bias of 0.6°-0.8°. Lui et al. (2021) show the WRF performance of Typhoon Hato in mean track bias within 1.5° compared with HKO's best track. Overall, the simulated tracks of simulations nudging horizontal wind above 500 hPa, which may reasonably capture the larger-scale circulation patterns, are close to the best tracks. Additionally,

nudging potential temperature, as well as the different initial times, CU, and MP schemes did not show remarkable differences in track bias. Compared with the other three TCs, Neogrui, which is generated in the SCS region, also shows large track improvement.





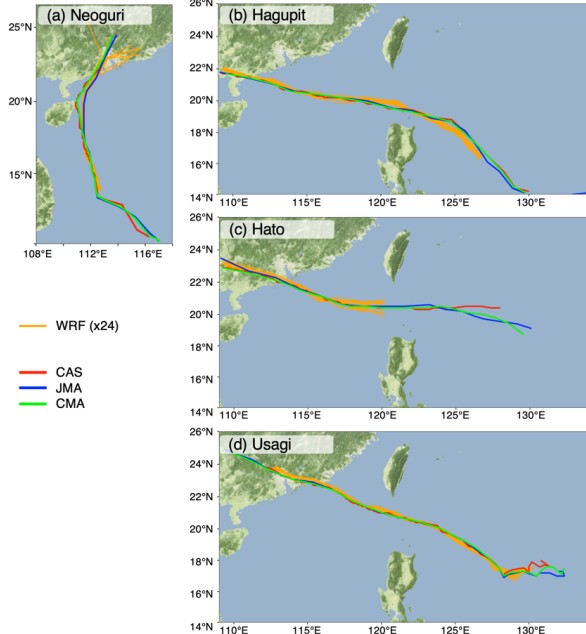

**Figure 2: Comparison of WRF simulation results and observed tracks of typhoons (a) Neoguri; (b) Hagupit; (c) Hato; (d) Usagi.**
**Red, green, and blue lines denote CAS, CMA, and JMA best tracks, respectively. The orange lines in each subplot denote the 24 experimental results.**

### 3.2 TC intensity

Figures 3a-d show observed and simulated MSLP for Typhoon Neoguri classified by the MP and CU configurations, spectral nudging settings, and initial times. Figures 3a-d show that compared with the MSLP of the three best tracks, the simulated 24
experiments could reproduce the MSLP changes with time. MSLP decreased from 12 UTC on April 16, reaching its lowest value around 970 hPa, and rising back from 12 UTC on April 17. Compared with the three best tracks, the lowest simulated value is around 5-10 hPa higher. Compared with the experiments by Potty et al (2012), MSLP is no less than 990 hPa for Neoguri, which is quite a weaker TC. The intensity bias already exists in the data used as the initial data (Mooney et al., 2019), partially attributed to its coarse resolution and impact on later simulations. In this study, we use 0.25° horizontal resolution
ERA5, although it relatively higher resolution compared with other reanalysis data, the bias is still inevitable.

As mentioned by Laux et al (2017), to avoid random errors from individual members, the ensemble mean has been widely used to balance out the errors. In this study we use the ensemble mean to show the different settings' influence on intensity, each set has at least 12 experiments which is more than the requirement of at least 10 cases.

Figure 3a shows that compared with W6, the TH ensemble mean has a relatively lower MSLP value, with differences reaching
around 1-4 hPa during the phase of typhoon intensification. When typhoons start weakening, the MSLP values align. Overall, at the intensification stage of Typhoon Neoguri, using the TH microphysics scheme resulted in a relatively higher intensity compared with W6.

Figure 3b shows that compared with CP, the KF ensemble mean has a lower MSLP value, with differences reaching up to 3-5 hPa during the phase of typhoon intensification. When the typhoons start weakening, the MSLP value from the CP ensemble
means remains higher in the first 12 hours of the weakening stage. KF has a higher intensity which is consistent with the previous simulations by Li et al (2018) using the 4 km horizontal resolution.

Figure 3c shows that, compared with PT+UV, the UV ensemble mean is mostly similar during the phase of typhoon intensification. However, while PT+UV starts weakening earlier, the UV ensemble mean remains at a high intensity even 12 h later, with differences reaching 3-5 hPa.





Figure 3d shows that compared with ST, TD and TS which start relatively early have a lower MSLP value, with differences
reaching 8-10 hPa during the phase of typhoon intensification. When the typhoons start weakening, the MSLP value from the
TD experiments' ensemble mean is still higher in the first 24 hours of the weakening stage.

Figure  S2a-d shows the sensitivity of the MWS for Typhoon Neoguri classified by the MP and CU configurations, spectral
nudging settings, and initial times. Although it can be seen from the figure that the different validation data inherit some

uncertainty because of wind averaging times, storm position, and intensity (Ying et al., 2014), marginally different in the MWS
of Typhoon Neoguri. Unlike MSLP, MWS shows a similar intensity compared with the observation which may be related to
the model representing the relationship between wind speed and sea level pressure. This phenomenon also exists in other
research mentioning that the drag coefficient controls the wind pressure relationship (WPR), and most of the simulated points
in the figure show higher MWS at the same MSLP compared with the reference (Kueh et al., 2019, Figure 4c).

Above all, for Typhoon Neoguri, in the typhoon intensification stage, using TH, KF, TD, and TS in the simulations resulted
in a lower MSLP and higher MWS compared with W6, CP, and ST. During the weakening phase of the typhoon, using UV in
the simulations leads to a higher MWS compared with PT+UV.

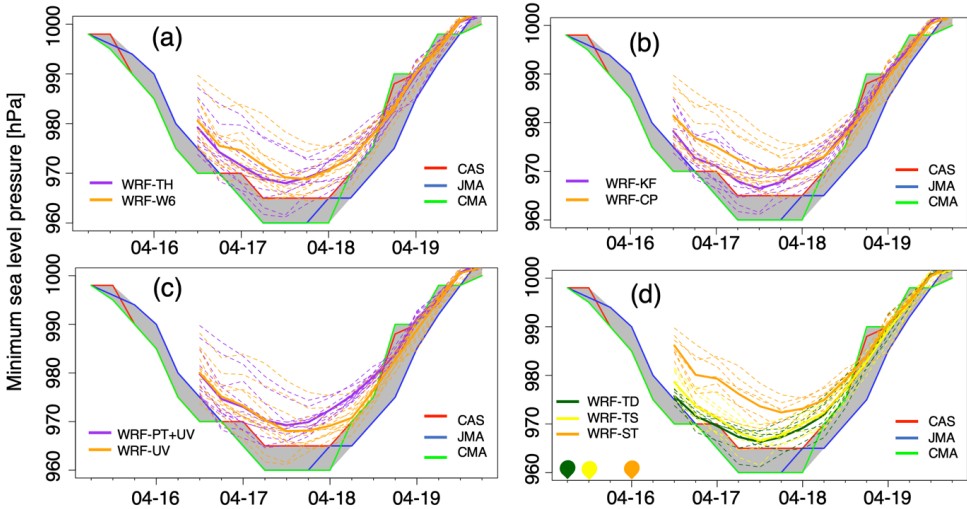

**Figure 3: Comparison of WRF simulation results and observed MSLP (unit: hPa) time series for Typhoon Neoguri during the**
**simulation period from 12 UTC April 16 to 00 UTC April 20, 2008. (a) for the MP schemes TH and W6; (b) for CU settings KF and**
**CP; and (c) for nudging settings PT+UV, UV; and (d) for different initial time TD, TS, and ST. The bold solid line represents the**
**ensemble mean of experiments.**

Figures 4a-d show the MSLP sensitivity for Typhoon Hagupit classified by MP and CU configurations, spectral nudging
settings, and initial times. Compared with Typhoon Neoguri, Typhoon Hagupit has a relatively longer intensification period

and higher intensity. Figures 4a-d show that compared with the three best tracks, the simulations could also reproduce the
MSLP changes with time, decreasing from 00 UTC on September 21 to around 958 hPa, keeping this intensity for around 24
h, and rising back from 12 UTC on September 23. Compared with the three best tracks, its lowest value is also around 5-10
hPa higher. Compared with the CP, nudging, and initial time settings, different MP schemes show similar values in MSLP.
For the CU settings, at the start time, the difference in MSLP is relatively small with ca. 1 hPa. With time, KF experiments

show a more intense intensification than CP. For nudging settings, nudging only horizontal wind has a relatively higher
intensity during the intensification period. For the initial time, Typhoon Hagupit has a relatively longer intensification period.
As for the late initial time, i.e., ST, we still allow for a spin-up time of 12-18 h for Neoguri and Hagupit, which is normally
used in previous research (e.g., Zhang et al., 2017). However, the simulations still show a weaker intensity. It means that not
only the time length should be considered but also the TC initial intensity. As mentioned before, the accuracy in representing





initial TC intensity is important for the subsequent simulation. Starting early, the absolute intensity bias may be relatively small and the model may capture more small-scale processes with relatively longer spin-up time, which likely benefit the intensification process, thus generating stronger TCs.

Figures S3a-d show the MWS sensitivity for Typhoon Hagupit classified by the MP and CU parameterization schemes, spectral nudging settings, and initial times. It shows that compared with the three best tracks, the simulated results could also reproduce

the MWS changes with time, increasing from 00 UTC on September 21 to around 50 m s$^{-1}$ and decreasing from 12 UTC on September 23. Figure  S3a-d shows that simulations in intensity are consistent with the MSLP of Hagupit.

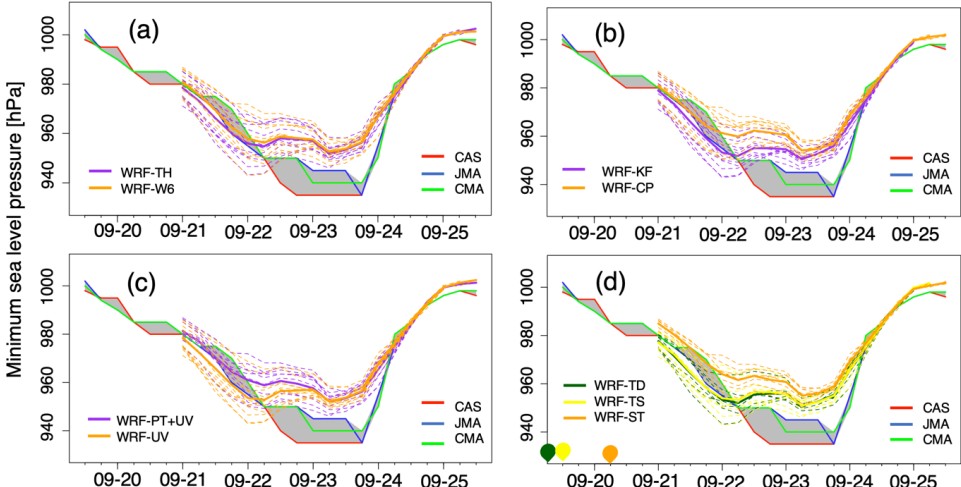

**Figure 4: Same as Figure 3, but for Typhoon Hagupit.**

Figures 5a-d show the sensitivity of Typhoon Hato in terms of MSLP classified by the MP and CU configurations, spectral

nudging settings, and initial times. Compared with Hagupit, Hato has a short intensification time, however, the overall intensity is stronger, indicating a more intense intensification process. The three best tracks show a large difference in the lowest MSLP, reaching 950 hPa for CMA, which is 20 hPa lower than JMA. At the start time, the intensities of observation and simulation are quite close to each other and WRF is also capable of capturing the following rapid intensification process. Most of the ensemble means of the different settings fall within the range of the three observed data sets at the lowest MSLP time. For

simulations of Typhoon Hato, compared to W6 (CP; PT+UV; TD), TH (KF; UV; TD or TS) shows a relatively lower MSLP during the typhoon intensification stage.

Figures S4a-d show the sensitivity of Typhoon Hato in terms of MWS classified by MP and CU configurations, spectral nudging settings, and initial times. The three best tracks also show a large difference at 00 UTC on August 23, reaching the highest MWS during the typhoon period. CMA indicates MWS of 45 m s$^{-1}$, which is 10 m s$^{-1}$ higher than JMA. For most of

the ensemble means of the different settings, the MWS is higher than the range of the three observed data sets during the intensification period. Compared to W6 (CP; PT+UV; TD), TH (KF; UV; TD or TS) shows a relatively higher MWS during the typhoon intensification stage.

Overall, for Typhoon Hato, using TH (KF; UV; TD or TS) settings resulted in a relatively lower MSLP and higher MWS and thus, a higher typhoon intensity. In a previous study, the average MSLP error was approximately 40 hPa without data

assimilation (Lu et al., 2019). The configurations used in our study are quite comparable to this previous research.

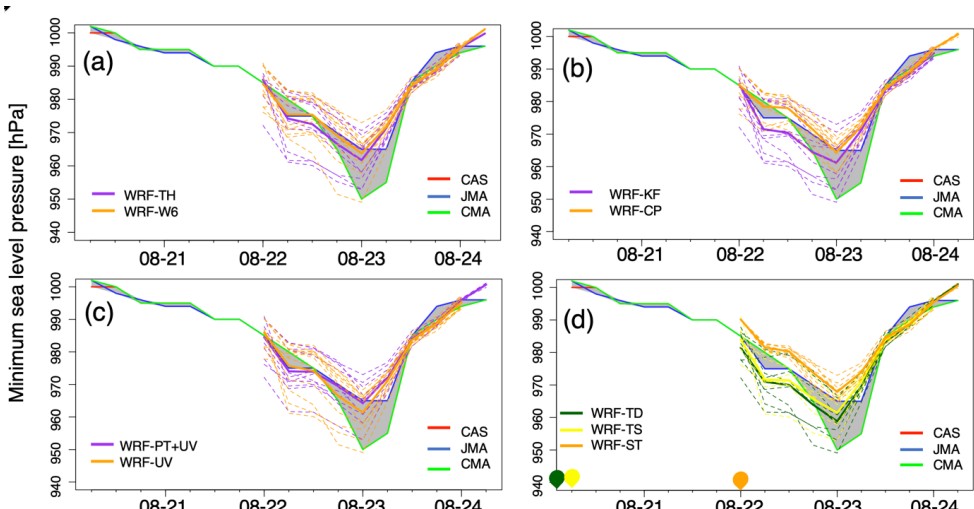

**Figure 5: Same as Figure 3, but for Typhoon Hato.**

Figures 6a-d show the sensitivity of Typhoon Usagi in terms of MSLP classified by the MP and CU configurations, spectral nudging settings, and initial times. Compared with the former three TCs, Usagi has the highest intensity and longest duration.

During the first several hours, the MSLP is around 10 hPa lower which means a higher intensity compared with the observation, unlike the three former typhoons. However, within the rapid intensification process, although WRF is, in general, capable of capturing this process, the intensity is much weaker than in the observation. At the time of the lowest MSLP, the bias reaches around 30 hPa. As shown in previous research by Gentry and Lackmann (2010), the intensities are likely to be underestimated. This especially applies to stronger TCs, because the coarse resolution may not fully resolve the intensification process, e.g.

mixing processes between the TC eye and eyewall that are influenced by small-scale processes like vortex Rossby waves and buoyant eyewall convection processes (Gentry and Lackmann, 2010). However, they also mention that 8 km is sufficient when aiming at realistically simulating the TC vortex.

For the MP scheme, TH and W6 show similar values in Figure 6a. For CU configurations, on the first day, KF has a lower MSLP than CP. However, the decrease of CP is quicker than KF. In the stage of steady MSLP, CP even has a slightly lower

value indicating a stronger typhoon than KF as shown in Figure 6b. Figures 6c-d show that PT+UV and ST show larger MSLP compared to UV, TD, and TS in the intensification period.

Figures S5a-d show the sensitivity of Typhoon Usagi in terms of MWS classified by the MP and CU configurations, spectral nudging settings, and initial times. At the start time, they are around 15 m s$^{-1}$ higher than observed. However, in the later intensification period, the MWS increases less rapidly compared to the observed data which is consistent with MSLP. For the

MP schemes, TH and W6 show similar values in Figure S5a. For CU settings, on the first day, KF has a higher MWS than CP. However, the decrease of CP is relatively quicker than KF. In the stage of steady MWS, CP even has a slightly higher value indicating a stronger typhoon than KF, as shown in Figure S5b. Figure S5c-d shows that PT+UV and ST show a lower MWS compared to UV, TD, and TS in the intensification period. Overall, for Typhoon Usagi, using KF (UV; TD or TS) settings resulted in a relatively lower MSLP and higher MWS, indicating a higher typhoon intensity during the intensification period.





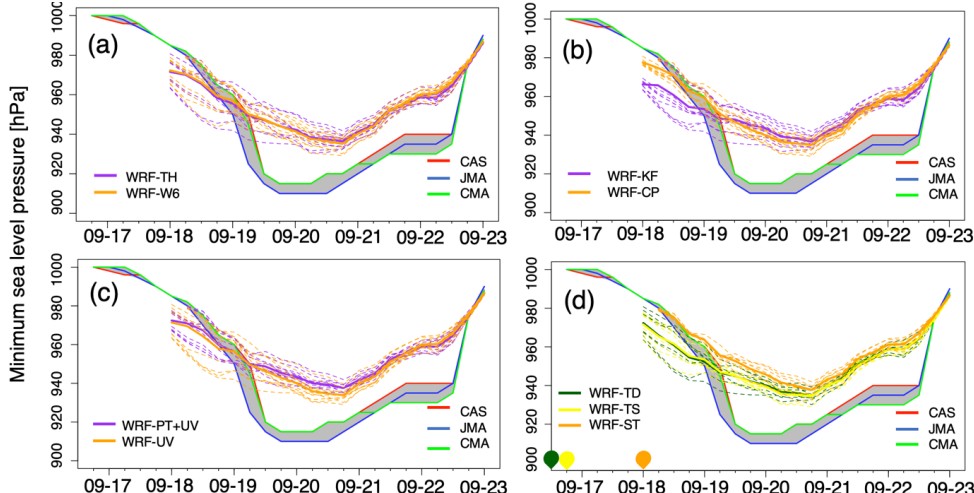


**Figure 6: Same as Figure 3, but for Typhoon Usagi.**

Above all, for TCs of different intensities, with different MP and CU configurations, spectral nudging settings, and initial times, in general, show consistency in terms of intensity. However, some differences remain. For four TCs, using KF (UV; TD or TS) settings resulted in a relatively lower MSLP and higher MWS and therefore a higher typhoon intensity. However,

compared with CU settings, spectral nudging settings, and initial times, the difference in TC intensity induced by the difference in MP schemes is relatively small during the intensification period, especially, for stronger TCs like Usagi. Compared with the CP, nudging and initial time settings, the TC cases show low sensitivity to the microphysical scheme which is consistent with previous research (Raktham et al., 2015).

Furthermore, we compared individual experiments to test the performance of the combination of different settings. Figures 7a-

d show the time average of bias in MSLP and MWS for Typhoon Neoguri, Hagupit, Hato, and Usagi from 24 individual experiments compared with CMA which uses 2 min mean MWS.

For Typhoon Neoguri, as shown in Figure 7a, TH_KF_(UV)_TS inherited less bias in MSLP of about 0.8 hPa. However, this setup shows an overestimated MWS during the simulation of around 6.5 m s$^{-1}$ which means the use of TH_KF_(UV)_TS combination could reproduce a relatively stronger typhoon compared with other combinations. The results are consistent with

the former ensemble mean results, for which the combination of TH, KF, UV, and TS show a relatively lower MSLP and higher MWS and therefore a higher typhoon intensity. However, the average intensity is stronger than the observation for Typhoon Neoguri. W6_KF_(PT+UV)_ST, W6_CP_(PT+UV)_ST, and W6_CP_(UV)_ST have less bias in MWS of around 2 m s$^{-1}$, however, they show an average overestimated MSLP during the simulation period of around 6.5 hPa which means this combination could reproduce a relatively weaker typhoon compared with most of the other combinations. The three

configurations have ST and W6 in common. Compared with former combinations, using TH_KF_(PT+UV)_TS, TH_KF_(UV)_ST, and W6_KF_(UV)_TD shows a relatively low combined error in MSLP and MWS. These three combinations have the KF setting in common.

Typhoon Hagupit, as shown in Figure 7b, TH_KF_(PT+UV)_TS and W6_KF_(PT+UV)_TS, shows less mean bias in MSLP and MWS. The commonality between the two combined schemes is using KF, PT+UV, and TS. The MP configurations show

less influence on the MSLP and MWS. Compared with the results of a previous simulation of Typhoon Hagupit (Sun et al., 2019), depicting mean MWS and MSLP simulation errors of 6 m s-1 and 11.3 hPa using JMA data as reference, the simulation bias in intensity is relatively low.

For Typhoon Hato, as shown in Figure 7c, W6_KF_(PT+UV)_TD, W6_KF_(UV)_TS, and W6_CP_(PT+UV)_TS have less mean bias compared with the other 21 experiments. Therefore, TS leads to better results. A previous study shows an MSLP of



around 950 hPa and therefore a bias of more than 20 hPa compared with the best track from HKO in the peak time (Lui et al., 2021).

For Typhoon Usagi, as shown in Figure 7d, because the typhoon has a very high intensity, unlike the former three TCs, 24 experiments show consistent underestimation in MSLP and MWS. TH_KF_(UV)_TD, TH_KF_(UV)_TS, and W6_KF_(UV)_TD show better results. Therefore, using KF, UV, TD or TS combinations will have better results for stronger

TCs. For Usagi, UV could reproduce a stronger typhoon and show better results, meaning nudging PT might inhibit the intensity of a stronger typhoon, consistent with the former research (Cha et al., 2011). However, Cha et al.(2011) used an intermittent spectral nudging method to reduce this negative effect. In our study, we believe nudging only horizontal wind speed could also maintain the positive effects of improving track bias and at the same time avoid partially negative effects induced by nudging more variables like potential temperature. The latter may influence intrinsic small-scale processes

reproduced by the WRF model, as the nudging technique impedes their development process because this information did not exist in the large-scale driving field as mentioned in the introduction.

Above all, when it comes to weaker typhoons, using KF, UV, TD or TS combinations results in an overestimated MWS. For Super Typhoon Usagi, the KF, UV, TD or TS combinations could reach relatively higher intensity, although there is still some underestimation. Besides, we also compared single experiments with JMA and CAS, as shown in Figure S6 and Figure S7.

Based on the ensemble mean, the error decreases. However, some error of around 7 hPa for MSLP and 3.37 m s$^{-1}$ for the 10-m wind speed (Di et al., 2019) remains. Compared with the previous research, our study ensemble error shows no more than 7 hPa for MSLP, and 3.37 m s$^{-1}$ for the MWS, which lies well within the error margins deemed acceptable in former research, with Super Typhoon Usagi being the exception.

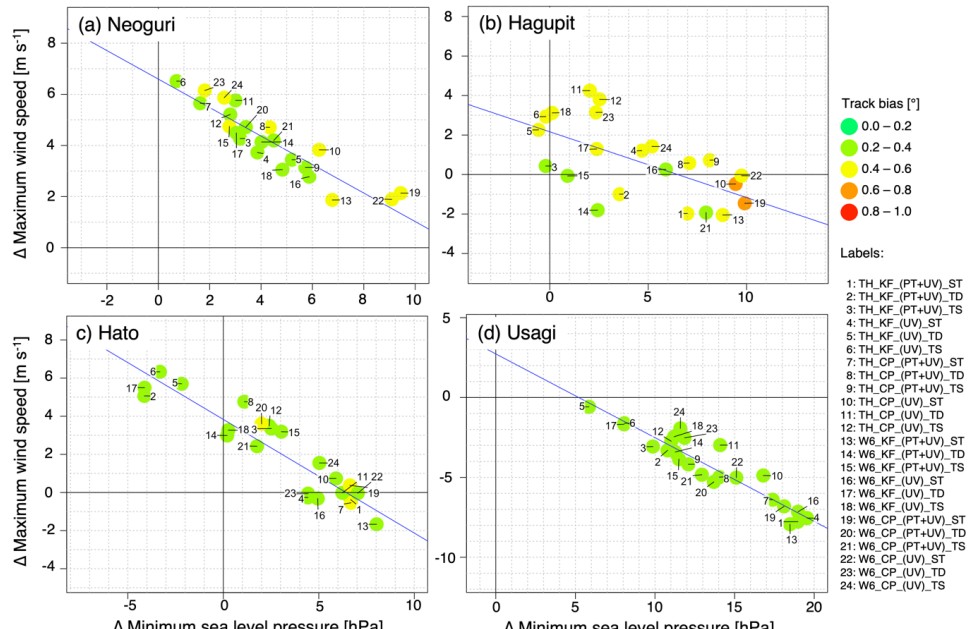

**Figure 7: Temporal average of bias for the track (colors), MSLP (x-axis), and MWS (y-axis) for (a) Typhoon Neoguri; (b) Typhoon Hagupit; (c) Typhoon Hato; (d) Typhoon Usagi compared with CMA.**

In Figure 8, we analyzed the simulated wind speed time series of the 24 different experiments in the grid cell closest to Zhuzilin station, which is located in the megacity Shenzhen within the PRD region and could be used for future evaluation. The aim is to study the sensitivity of strong wind speeds to different configurations in coastal cities which may cause storm surges and

other disasters.





For Typhoon Neoguri, the wind speed of the TH scheme has approximately a 1 m s$^{-1}$ higher wind speed compared with W6. For CU, in the strengthening stage of the typhoon, the wind speed of the KF scheme is relatively strong, but decreases rapidly after 12 UTC, while CP reaches its peak at 13 UTC on April 19th. Then, the wind speed decreases, while CP remains 2 m s$^{-1}$ higher than KF. Nudging only UV resulted in a relatively higher wind speed. For different initial times, ST showed a relatively

higher wind speed. For Typhoon Hagupit, all simulations in Figure 8b indicate rising wind speeds, reaching the peak at around 12 UTC, and consistently decreasing afterward. Nudging PT+UV shows a relatively higher wind speed compared with UV during the wind speed reduction period. For Typhoon Hato, ST is relatively weaker compared with early initial time simulations which might be related to the radial extent of winds of approximately 15 m s$^{-1}$ being broader in TD and TS. Because the simulated location of the typhoon is similar as shown in Figure 2 and Figure 7. For Typhoon Usagi, the wind

speed is relatively higher for PT+UV compared with UV at the starting time. In the other period, the 24 experiments are quite close to each other. Above all, the simulated wind speed shows deviations among different experiments, however, the trend is similar and the wind speed average difference is within 2 m s$^{-1}$. The simulation results still need to be compared with meteorological station data.

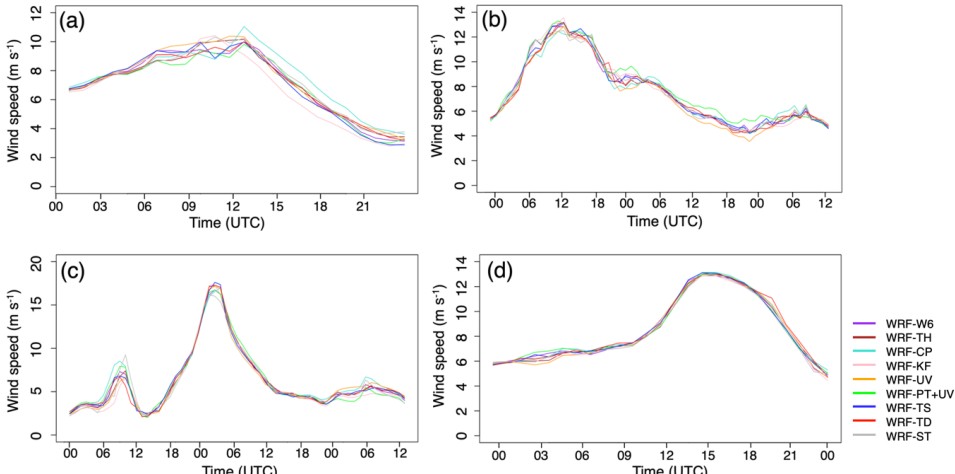

**Figure 8: Intercomparison of simulated wind speed (m s$^{-1}$) time series (24 WRF experiments) at Zhuzilin station in the PRD region for (a) Typhoon Neoguri; (b) Typhoon Hagupit; (c) Typhoon Hato; (d) Typhoon Usagi.**

### 3.3 TC precipitation

The rainfall amount is influenced by the convection process so it could also be used as an indicator of the convection process and TC strength (Gao and Chiu, 2010). In this study, we compare accumulated rainfall in the typhoon period from different

MP and CU configurations, spectral nudging settings, and initial times and compare it with GPM data as shown in Figure 9 for Typhoon Neoguri. The spatial pattern of accumulated rainfall is highly related to the tracks according to Figure 2. All the simulations show similar spatial characteristics, which significantly differ from the No-nudging simulations (Figure S1), whose track and rainfall region is shifted to the west (not shown). Compared with GPM rainfall data, the spatial pattern and magnitude of typhoon rainfall are similar, indicating that the simulation results are reasonable.

The inspected MP schemes use different moisture species and physical processes involved in the phase changes to represent the formation of rainfall (Xu et al., 2023). Compared with W6, using TH leads to heavier rainfall when passing the ocean around 15 °N, as shown in Figure 9a-b. As shown in Figure 3a, the intensity is also higher in TH compared with W6, likely related to more rainfall releasing more latent heat flux into mass, which could support the vertical process and transfer more air into the upper level, further decreasing the MSLP.



As for CU settings, before passing 15 °N, KF has a larger rainfall than CP, consistent with a stronger TC intensity in Fig 3b. Later, CP has a relatively broader rainfall region compared with KF. Also, the coastal region shows more rainfall. As mentioned before, the KF scheme will induce more convective processes compared with CP which is insufficient to faithfully capture the entire range of convective motions (Bryan et al., 2003), inducing more rainfall as well as a lower MSLP. Compared with PT+UV, UV settings have a broader rainfall area with precipitation sums above 300 mm. When the TC passes these regions,

TC intensities are higher in UV compared with PT+UV, as shown in Figure 3c. Furthermore, in the coastal region of Guangdong, UV shows around 100-120 mm of accumulated rainfall, whereas PT+UV shows around 80-100 mm. Unlike the formal physical process, nudging is a technique that impacts simulated large-scale fields. Nudging PT+UV inhibits the TC intensity more. For different initial times, before passing 15 °N, TD and TS have a larger rainfall amount compared with ST which is also consistent with Figure 3d, because starting early may give the small-scale process more time to develop in WRF.

Above all, the simulated TC accumulated rainfall is highly related to the simulated track and intensity.

We further evaluated the simulated rainfall time series of the grid cell closest to Zhuzilin station, as shown in Figure 10. In Figure 10a, WRF could capture the typhoon-induced rainfall process. However, while there are two peak times in GPM, the model can only simulate a single peak of precipitation. As for peak hourly rainfall rate, the range of 24 experiments is around 16-31 mm, and GPM is 25 mm. As for station rainfall, it still needs to be compared with station data.

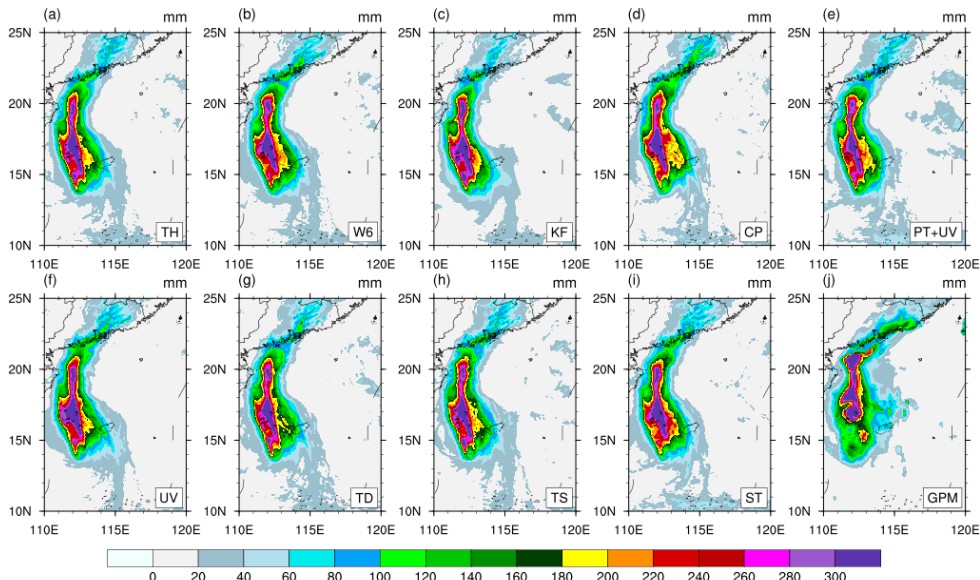


**Figure 9: Spatial patterns of accumulated rainfall (mm) for Typhoon Neoguri from 1200 UTC on 16 April 2008 to 0000 UTC on 20 April 2008 for (a-c) TH, W6; (d-e) KF, CP; (f-g) PT+UV, UV; (h-i) TD, TS, ST, and (j) GPM.**

For Typhoon Hagupit, as shown in Figure S8, compared with GPM, the accumulated typhoon rainfall spatial pattern and magnitude are similar. However, the range of the maximum precipitation area is relatively narrow. As the typhoon made

landfall to the west of the PRD, the rainfall at the specific grid cell is relatively small compared with other typhoon cases. The simulations show an overestimated 60-80 mm precipitation, whereas GPM indicates 20-40 mm (Figure S8), which might be related to a slight eastward shift of the landfall region in the simulation, as shown in Figure 2b. In Figure 10b, WRF could capture the typhoon-induced rainfall process, however, the peak time of precipitation exhibits a lag and the magnitude is larger than GPM. The magnitude of simulated rainfall is relatively larger which might also be related to the eastward shift of the TC

track.





For Typhoon Hato, as shown in Figure S9, like Hagupit, GPM has a relatively broader rainfall region compared with the simulations in Figure S9. As for coastal regions, some overestimated rainfall exists compared with GPM. As shown in Figure 10c, there are two peak rainfall times. For the first peak time, the simulations consistently underestimate the rainfall rate, however, in the second peak time, the rainfall is overestimated. Compared with the ground observation from Lu et al. (2019),

the simulation results prove to be reasonable.

As shown in Figure S10, Typhoon Usagi shows a similarity between simulated rainfall and GPM data in the magnitude and range of the maximum precipitation area when passing the ocean. When arriving at the coastal regions, the simulated results show some underestimation in the PRD region. As for the grid cell closest to Zhuzilin station, the simulated TC has a relatively longer continuous rainfall time, however, the hourly rainfall rate has been shown to underestimate the value.

The rainfall intensity in the ocean shows some consistency with TC intensity, however, when it comes to the coastal regions, which inhabit a large population, there is no obvious relationship with TC intensity. There exists a difference in rainfall rate from different configurations which might be induced by the influence of the size of the spiral band which should be further studied.

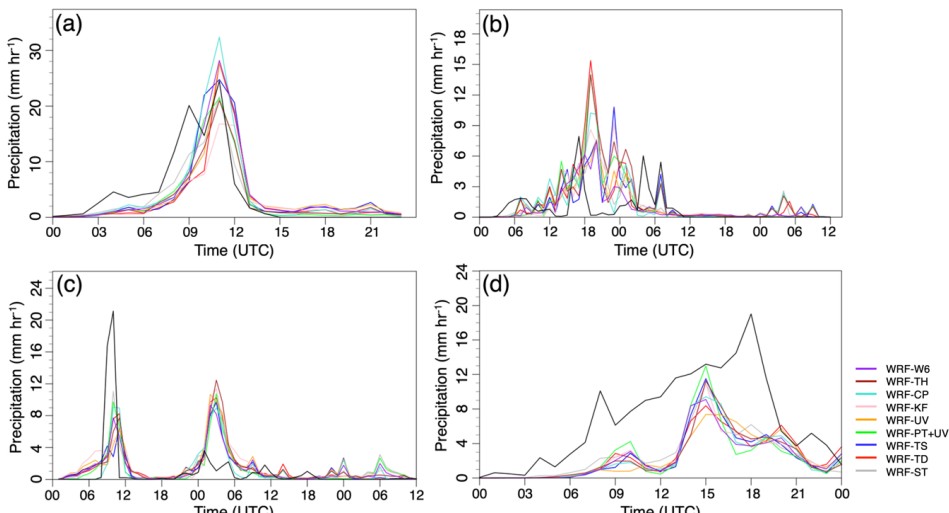

**Figure 10: Intercomparison of simulated hourly rainfall (mm h⁻¹) time series (24 WRF experiments) at Zhuzilin station in the PRD region for (a) Typhoon Neoguri; (b) Typhoon Hagupit; (c) Typhoon Hato; (d) Typhoon Usagi.**

### 3.4 Physical changes

Based on Sun et al. (2019), latent heat flux is the main heat source for TC development and they also speculate that the simulated latent heat flux is related to the simulated TC intensity. Convective processes, by influencing sensible and latent

heat and momentum transport, affect the vertical structures of the atmospheric temperature and humidity fields. In our research, we further investigate whether CU settings induced differences in latent heat flux (Figure 11). In Figure 11a, for Typhoon Neoguri, compared with CP, KF consistently shows a relatively larger latent heat flux within the track of TCs. As shown in Figures 11b-d, the same phenomenon also occurred along the paths of the other three typhoons. The absolute value of latent heat flux is around 300 W m⁻², and the difference is approximately 20-30 W m⁻², which is about 10 %.

We further analyzed the vertical structures of the equivalent potential temperature θe which is a variable that combines temperature, pressure, and humidity, and is a thermodynamic parameter used to assess the moist static energy content of an air mass, which is closely related to the TC intensity (Ma et al., 2013). The field could show that a more intense TC is related to the warmer core of the TC because of two processes: firstly, higher latent heat flux between ocean and air induces more air ascending in the eyewall and releases more latent heat as vapor within rising air parcels condenses. The second process is



related to subsidence in the eye of the storm causing further warming in the eye through compressional heating potentially lowering the surface pressure (Gentry and Lackmann, 2010). θe is also used as a criterion for convective instability (CI).  As shown in Figure 12a-d, KF has a higher θe in all layers and CI reaches higher layers compared with CP, allowing more heat to be transported from the lower to the upper atmosphere. This results in a significantly warmer core structure of the typhoon. Based on these results, we speculate that the KF scheme generates stronger ocean–TC interactions which stimulate TC

development. Compared with CP, KF then transports heat energy to the upper layer, resulting in differences in the simulated convective process, further generating heavier precipitation (Figure 9c-d). The rainfall leads to variations in latent heat release, which, in turn, impacts the convective process and shows the stronger intensity of typhoons, which is consistent with the wind-induced heat exchange (WISHE) mechanism (Emanuel, 1986).

As mentioned by Cha et al (2011), nudging might weaken TC intensity. In this study, we also investigated the impact of

nudging on the TC intensity. As shown in Figure S11, compared with PT+UV, UV and No-nudging shows a higher θe at all layers and higher CI height. Nudging PT+UV results in a decrease in the simulated typhoon intensities. However, exclusively nudging horizontal wind can alleviate the inhibiting effect. Nudging horizontal wind could have a positive effect on reducing the track distance error, however, nudging too many variables may not lead to further improvements on the tracks and even inhibit the TC development during the intensification period.

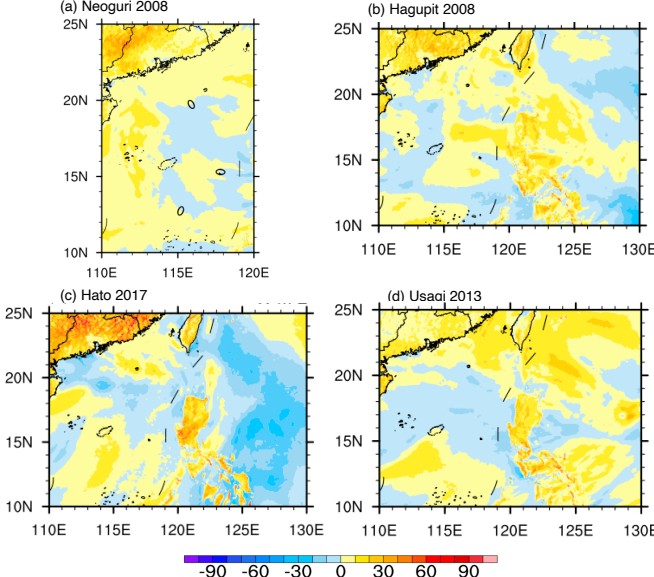


**Figure 11: Spatial patterns of simulated latent heat flux (w m⁻²) average difference between KF and CP for typhoons (a) Neoguri; (b) Hagupit; (c) Hato; (d) Usagi.**





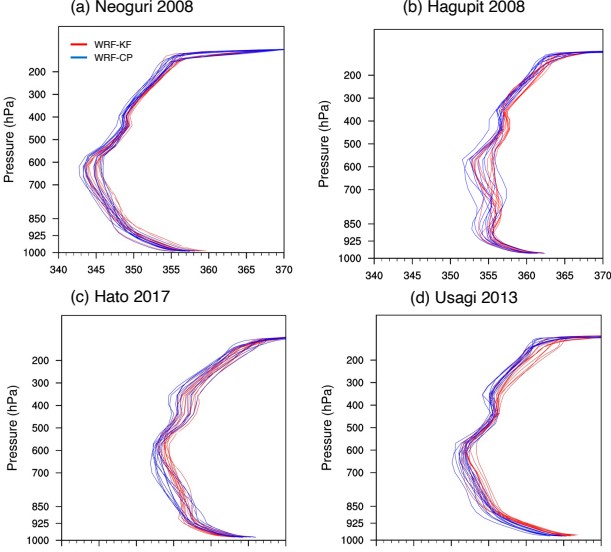

**Figure 12: Equivalent potential temperature (θ$_e$) vertical distribution between KF (red) and CP (blue) for (a) Typhoon Neoguri on Apr 17th 00 UTC; (b) Hagupit on Sep 22nd 12 UTC; (c) Hato on Aug 23rd 00 UTC; (d) Usagi on Sep 18th 12 UTC.**

## 4 Summary and Conclusion

In this study, we use the WRFV4.3 model to conduct numerical experiments on the sensitivity of Typhoon Neoguri, Hagupit, Hato, and Usagi with different MP, CU, nudging, and initial time settings. These four typhoons were selected as instances of different TC categories (based on the Saffir-Simpson scale) causing compound flood events in the PRD region.

The results led to the following conclusions:

1. For the track simulations, nudging only horizontal wind could reasonably capture the large-scale circulation patterns, leading to simulated tracks quite close to the best track reference data sets. Like the other three TCs, Neogrui, which is solely generated in the SCS region, also shows a large track improvement.

2. For intensity, the four selected TCs show consistency: compared with CP (PT+UV; ST), using KF (UV; TD or TS) settings shows a relatively lower MSLP and higher MWS indicating a higher typhoon intensity. Compared with the CU, nudging, and initial times, the TC cases show low sensitivity to the MP which is consistent with former research. Physical changes between CU configurations show that, from the perspective of energy, KF has a relatively larger latent heat flux between ocean and air which allows for an increased energy provision to the TC system. Moreover, a higher equivalent potential temperature indicates that the larger moist static energy in the TC system, which may partly transfer to kinetic energy, may induce a stronger TC during the intensification period. Therefore, especially for the simulation of super typhoons, this study proposes to use a KF, UV, TD, or TS configuration combination, although the intensities are still underestimated.

3. Although the different simulation results show a large difference in MWS and SLP, the differences are not very pronounced on land in the coastal regions.

There remain limitations of this study. Due to the lack of observation stations, the simulated results around coastal regions still need to be validated. Besides, the WRF performance could be improved with data assimilation when more observation data becomes available. Furthermore, there still remains a variety of uninvestigated CU and MP, PBL schemes, and different surface flux options that impact TCs.



Future work will apply this model configurations with different CMIP6 projections results using the pseudo global warming (PGW) approach to improve our understanding of potential future changes in TCs and also utilize these results for hydrodynamical models such as the Delft3D, from which the cities' disaster management and defense could potentially benefit.

**Data availability**

The data used in this study can be accessed by contacting the first author. The WRF results can be accessed upon request. ERA5 data was downloaded from the European Centre for Medium-Range Weather Forecasts (ECMWF), Copernicus

Climate Change Service (C3S) at Climate Data Store (CDS; https://cds.climate.copernicus.eu/) to derive the WRF. GPM IMERG Final Precipitation L3 Half Hourly 0.1 degree x 0.1 degree V06 (GPM_3IMERGHH) data was downloaded from the EARTHDATA database available at https://disc.gsfc.nasa.gov/datasets?keywords=GPM&page=1 and three TC track data are from Oceanographic Data Center, Chinese Academy of Sciences (CASODC) (http://msdc.qdio.ac.cn/), the China Meteorological Administration (CMA)(https://tcdata.typhoon.org.cn/), the World Meteorological Organization (WMO)

Regional Specialized Meteorological Center in Tokyo, Japan (JMA)(https://www.jma.go.jp/), used to verify the simulation results. WRF model is available at https://www2.mmm.ucar.edu/wrf/users/download/get_source.html.

**Supplement**

The supplementary material is added to this article.


**Author contributions**

The study's conceptualization was a collaborative effort involving QS, PO, and PL. QS took on the responsibilities of experiment execution and initial manuscript drafting. Subsequently, both QS and PO were involved in the comprehensive tasks of data analysis, visualization, and validation. PO, PL, LS, and ZT contributed significantly to the meticulous review

and editing process. PO also played a pivotal role in the development of the typhoon tracking algorithm. JW provided essential support for the model operation. PL and HK provided invaluable supervision throughout the study. PL undertook project administration, while HK facilitated the acquisition of funding. It is important to emphasize that all authors actively participated in the interpretation of the results and made substantial contributions to the manuscript's refinement.

**Competing interests**

The authors declare that they have no known competing financial interests or personal relationships that could have appeared to influence the work reported in this paper.

**Disclaimer**

**Special issue statement**

This article is part of the special issue "Attributing and quantifying the risk of hydrometeorological extreme events in urban environments". It is not associated with a conference.

**Acknowledgments**

We thank the following institutions for providing data: Oceanographic Data Center, Chinese Academy of Sciences (CASODC) (http://msdc.qdio.ac.cn), the China Meteorological Administration (CMA)(https://tcdata.typhoon.org.cn/), the World



Meteorological Organization (WMO) Regional Specialized Meteorological Center in Tokyo, Japan (JMA)(https://www.jma.go.jp/).

**Financial support**

This study was conducted in the framework of the Sino-German project Mitigating the Risk of Compound Extreme Flooding Events MitRiskFlood, funded by MOST (grant number 2019YFE0124800) and the German Ministry of Education and
Research (BMBF) (grant number 01LP2005A). Qi Sun is supported financially by the Chinese Scholarship Council (CSC).

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
