# Peer review of "Key ingredients in regional climate modelling for improving the representation of typhoon tracks and intensities"

_Hydrology and Earth System Sciences, 2023_

## Referee Comment (RC1)

Review of "Key ingredients in regional climate modeling for improving the representation of typhoon tracks and intensities"

By Sun et al.

Submitted to HESS Manuscirpt # hess-2023-222

Recommendation: Minor revision

Summary

This study investigates the influence of different combinations of model parameterization, nudging, and initial conditions on typhoon simulations using the WRF model, focusing particularly on the PRD region. Four typhoon cases within the region were simulated, and recommendations regarding the WRF model configuration in the area are provided. In comparison to the heavily cited work of Sun et al. (2019) in this manuscript, the emphasis of this study lies more on the model configuration, with less emphasis on comparing the results to observations as extensively done by Sun et al. (2019). I think this aspect could be improved if similar observation is available to the authors. However, with the substantial number of simulations conducted in this study, the results in this work appear robust. I have only minor comments for the authors to consider. It is recommended to pursue publication if the comments below can be satisfactorily addressed.

Minor comments:

L23: "Cu" should be "CU"

L40-45: the authors might want to cite another recent study below highlighting a potential positive trend in major TC landfall.

> Wang, S. and Toumi, R., 2022. More tropical cyclones are striking coasts with major intensities at landfall. Scientific reports, 12(1), p.5236.

L79: "Sun et a.," should be "Sun et al."

L164: It appears the scientific question of the study can be further polished and specified. The authors tried some CU and MP schemes, but not all. Four typhoons were simulated, but, again, not all the typhoons in the region. Therefore it would be difficult to answer the question the authors put there for themselves, i.e., what is the best combination to simulation TCs in the PRD with WRF. Instead of finding the best combination, which is almost undoable with the current model setup, I wonder if it makes more sense to explore why one combination of schemes is better than the rest that were tested in the authors' simulations. Here I'm not suggesting more

simulations, rather, I wonder if it would be helpful to rephase certain sentences in the manuscript for better clarity.

L175: Again, I'm not asking the authors to conduct more simulations to increase the grid spacing, but it might be worth explaining why the resolution is only 5 km. Nowadays even higher grid spacing has been applied with some operational setup with ensemble simulations.

Table 1. I'm not sure "Convection-permitting (CP)" is the best way in defining the suite of simulations with no cumulus scheme. As the authors mentioned, 5 km "is within the gray zone" (L185), the 5-km grid spacing may not have convection permitted.

Figure 1. Please mark the PRD region directly with a box in the plot.

Figure 2. Please improve the resolution of the figure.

L526: please define $\theta_e$ first.

---

## Author Comment (AC1)

Response to Reviewers' Feedback
Original comment in black, *response in blue and italics*

*We thank the reviewers for their time and the thorough examination. Their feedback helped to improve the quality of the manuscript. The main changes that have been made to the manuscript include the following:*

- *We enhanced the image quality and added auxiliary lines to better explain the results.*
- *We modified language and expression to make the statement clearer and more accurate.*
- *We included additional relevant literature to assist readers in better understanding the study.*

*Please note that all information concerning pages and lines refers to the revised clean version. Below, we address each of the reviewer's comments in detail:*

**Reviewer #1:** Summary. This study investigates the influence of different combinations of model parameterization, nudging, and initial conditions on typhoon simulations using the WRF model, focusing particularly on the PRD region. Four typhoon cases within the region were simulated, and recommendations regarding the WRF model configuration in the area are provided. In comparison to the heavily cited work of Sun et al. (2019) in this manuscript, the emphasis of this study lies more on the model configuration, with less emphasis on comparing the results to observations as extensively done by Sun et al. (2019). I think this aspect could be improved if similar observation is available to the authors. However, with the substantial number of simulations conducted in this study, the results in this work appear robust. I have only minor comments for the authors to consider. It is recommended to pursue publication if the comments below can be satisfactorily addressed.
*Thank you very much for the generally positive statements about our study.*

Minor comments:
L23: "Cu" should be "CU"
*We agree with the reviewer about the inconsistency regarding abbreviations and have changed all expressions concerning these abbreviations.*

L40-45: the authors might want to cite another recent study below highlighting a potential positive trend in major TC landfall.
Wang, S. and Toumi, R., 2022. More tropical cyclones are striking coasts with major intensities at landfall. Scientific reports, 12(1), p.5236.
*Thanks for the recommendation. We included this publication and cited it in the introduction (Line 43, revised clean version).*

L79: "Sun et a.," should be "Sun et al."
*Thank you, we corrected this typo.*

L164: It appears the scientific question of the study can be further polished and specified. The authors tried some CU and MP schemes, but not all. Four typhoons were simulated, but, again, not all the typhoons in the region. Therefore, it would be difficult to answer the question the authors put there for themselves, i.e., what is the best combination to simulate TCs in the PRD with WRF? Instead of finding the best combination, which is almost undoable with the current model setup, I wonder if it makes more sense to explore why one combination of schemes is better than the rest that was tested in the authors' simulations. Here I'm not suggesting more simulations, rather, I wonder if it would be helpful to rephrase certain sentences in the manuscript for better clarity.
*We agree with the reviewer that the scientific question of the study can be further polished and specified and we have changed it. We followed the suggestion of the reviewer and tried to explore why one combination of schemes performs better than the rest for specific TCs in this region. We modified the expression (Line 167-171, revised clean version), as follows:" The main objective of this study is to analyse the uncertainties from different combinations of schemes in WRF to represent TC Neoguri*

*(2008), Hagupit (2008), Hato (2017), and Usagi (2013) affecting the PRD region with different intensities and genesis locations. More specifically, it is analysed (i) how sensitive the typhoons belonging to specific intensity categories or genesis locations are to two CU (KF and CuOFF) settings and two MP (WSM6 and Thompson) parameterization schemes, three initialization times (TD, TS, ST), and two spectral nudging variables (UV, PT+UV). (ii) It is explored why the chosen combination of schemes is better than the rest from a thermodynamic perspective."*

L175: Again, I'm not asking the authors to conduct more simulations to increase the grid spacing, but it might be worth explaining why the resolution is only 5 km. Nowadays even higher grid spacing has been applied with some operational setup with ensemble simulations.

*We agree with the reviewer that nowadays even higher grid spacing has been applied to some operational setups with ensemble simulations. In this study, however, a balance between computational efficiency and model accuracy is achieved by using a horizontal resolution of 5 km for typhoon simulations. This domain selection considered studies like Sun et al. (2013) and Gentry and Lackmann (2010), which indicate less impact on typhoon intensity within a certain resolution threshold. Furthermore, similar horizontal resolutions have been applied by Gutmann et al. (2018) and Delfino et al. (2022). The references are listed below.*

Table 1. I'm not sure "convection-permitting (CP)" is the best way in defining the suite of simulations with no cumulus scheme. As the authors mentioned, 5 km "is within the gray zone" (L185), the 5-km grid spacing may not have convection permitted.

*Thanks for the suggestions. We modified the name into CuOFF, meaning that in this experiment we switched off the cumulus scheme.*

Figure 1. Please mark the PRD region directly with a box in the plot.
*Thanks for the suggestion. The figure is updated accordingly in the revised version of the manuscript.*

Figure 2. Please improve the resolution of the figure.
*Thanks for the suggestion. We will enhance the resolution of the figure to ensure it is more visually clear. The figure with improved resolution will be included in the revised version of the manuscript.*

L526: please define θe first.
*Thanks for the suggestion, we define $\vartheta_e$ first (Line 244, revised clean version) which could be beneficial for the reader's understanding.*

*References*
*Delfino, R. J., Bagtasa, G., Hodges, K., and Vidale, P. L.: Sensitivity of simulating Typhoon Haiyan (2013) using WRF: the role of cumulus convection, surface flux parameterizations, spectral nudging, and initial and boundary conditions, Nat. Hazards Earth Syst. Sci., 22, 3285–3307, https://doi.org/10.5194/nhess-22-3285-2022, 2022.*

*Gentry, M. S. and Lackmann, G. M.: Sensitivity of Simulated Tropical Cyclone Structure and Intensity to Horizontal Resolution, Mon. Weather Rev., 138, 688–704, https://doi.org/10.1175/2009MWR2976.1, 2010.*

*Gutmann, E. D., Rasmussen, R. M., Liu, C., Ikeda, K., Bruyere, C. L., Done, J. M., Garrè, L., Friis-Hansen, P., and Veldore, V.: Changes in Hurricanes from a 13-Yr Convection-Permitting Pseudo–Global Warming Simulation, J. Clim., 31, 3643–3657, https://doi.org/10.1175/JCLI-D-17-0391.1, 2018.*

*Sun, Y., Yi, L., Zhong, Z., Hu, Y., and Ha, Y.: Dependence of model convergence on horizontal resolution and convective parameterization in simulations of a tropical cyclone at gray-zone resolutions: RESOLUTION AND CONVECTION IN TC SIMULATION, J. Geophys. Res. Atmospheres, 118, 7715–7732, https://doi.org/10.1002/jgrd.50606, 2013.*

---

## Author Comment (AC2)

Response to Reviewers' Feedback
Original comment in black, *response in blue and italics*

*We thank the reviewers for their time and the thorough examination. Their feedback helped to improve the quality of the manuscript. The main changes that have been made to the manuscript include the following:*

- *We enhanced the image quality and added auxiliary lines to better explain the results.*
- *We modified language and expression to make the statement clearer and more accurate.*
- *We included additional relevant literature to assist readers in better understanding the study.*

*Please note that all information concerning pages and lines refers to the revised clean version. Below, we address each of the reviewer's comments in detail:*

**Reviewer #2:**

The authors did a good job in presenting interesting analyses and results to the scientific community. However, as there are a few things requiring improvement in the manuscript at this point, it is suggested that the article should go through minor revisions and thorough English editing before accepting and publishing. Specific review comments are provided to the authors as follows.

Lines 046 – 101: Although plenty of studies have been reviewed in the INTRODUCTION part, the authors are encouraged to review and include some more relevant works published recently. For example, Hsu et al. (2023; https://doi.org/10.5194/nhess-2023-49) analysed how storm characteristics (e.g., TC path, intensity, heading direction, and translation speed) affect storm surge and wave runup along the coast during three historical Atlantic hurricanes. Zhang et al. (2023; https://doi.org/10.1016/j.oceaneng.2023.113977) used a storm surge model to study the wave and storm characteristics as well as how they responded to land reclamations in the Pearl River Estuary between 1990 and 2020. Furthermore, given the background of global warming, the effects of typhoon intensity increase on changes in waves and storm surges brought on by reclamations were also demonstrated.
*Thanks for the recommendations. We learned a lot from these publications and cited them in the introduction (Line 55 and Line 59, revised clean version).*

Lines 114 – 116: The authors are encouraged to rewrite this sentence to make it clearer for the readers to follow.
*Thanks for the suggestions. We rewrote this sentence to make it clearer for the readers (Line 115-118, revised clean version).*

Lines 187 – 189: First, what the authors wanted to express here may be confusing. Did the authors mean "For spectral nudging, we investigated the nudged horizontal wind above 500 hPa, the nudged horizontal wind, and the nudged potential temperature."? The authors are encouraged to rephrase and rewrite the statement. In addition, did the authors mean "For the initial condition, we define the starting time of the simulation based on the TC intensity with an attempt to assess which initial time produces the most accurate results."? The authors are encouraged to rephrase and rewrite these statements to make it clearer for the readers.
*Thanks for the suggestions. We rephrased and rewrote these statements to make them clearer for the readers (Line 191-194, revised clean version).*

Lines 201 – 202: "Propagating westward and passing over the Sulu Sea and then moving gradually to the northwest." This sentence does not have a subject or verb. The authors are encouraged to rephrase and rewrite the statement.
*Thanks for the suggestions. We corrected this (Line 207, revised clean version).*

Lines 204 – 204: The authors are encouraged to revise "The difference between Typhoon Neoguri and the other considered storms is …" to "One of the main differences between Typhoon Neoguri and the other considered storms is …" because there are numerous differences between storms (e.g., path, intensity, radius of maximum wind).

*Thanks for the suggestions. We rewrote this sentence to make it clearer for the readers (Line 210, revised clean version).*

Lines 210 – 212: The authors are encouraged to indicate the UTC times for each relevant instant. For example, the authors are encouraged to rewrite the statement like "At 2017-08-20-00 UTC, Typhoon Hato originated over the sea east of the Philippines … Three hours later (i.e., 2017-08-20-03 UTC), it attained Super Typhoon intensity …". (Note that the times here are only examples. The authors are encouraged to indicate the correct times similarly.) This comment also applies to the statements regarding all the considered typhoons (e.g., lines 201—203, lines 214—216).

*Thanks for the suggestion. We revised the relevant sentences, such as lines 206, 208, 212 and 213 to incorporate the UTC times because of the importance of the precise time references for clarity and accuracy in our study.*

Lines 237 – 237: While the full name of MSLP was given in line 126, the authors are encouraged to indicate the full name of the abbreviation "SLP" here as this is the first time it appeared in the manuscript.

*Thanks for the suggestion. we provided the full name of 'SLP' - Sea Level Pressure - at its first occurrence in the revised version of the manuscript to ensure comprehensibility (Line 253, revised clean version).*

Lines 255 – 255: The authors are encouraged to express the statement more precisely. For example, this sentence may be clearer to be revised as "Typhoon track prediction could be significantly enhanced by nudging horizontal wind, which influences large-scale circulation patterns and steering flow."

*Thanks for the suggestion. We agreed the revised sentence would make the statement more precise (Line 270-271, revised clean version).*

Lines 262 – 262: The authors are encouraged to use an en-dash to describe any range of numbers/values instead of a hyphen. This also applies to all the other places describing a range of numbers/values in the document (e.g., lines 264, 268, 271, 272, 284, 287, 295, 298, 304, 306, 308, 323, 325, 332, 338, 344, 363, 375, 382, 394, 476, 484, 491, 518, 519, and 527).

*Thank you for the valuable suggestion regarding the use of an en-dash for numerical ranges. We will ensure that this change is consistently applied to all numerical ranges throughout the revised version.*

Lines 286 – 287: The authors are encouraged to rewrite this statement to make the subjects clearer. For example, "Compared to the best tracks of the other three storms, the lowest simulated MSLP during Typhoon Neoguri is around 5–10 hPa higher."

*Thanks for the suggestion. We rewrote the statement to make the subjects clearer (Line 301, revised clean version).*

Lines 289 – 290: The authors are encouraged to revise the sentence as "In this study, we use ERA5 with a 0.25 horizontal resolution, the bias is still inevitable despite its relatively high resolution compared to other reanalysis datasets." There are numerous instances of such grammar problems throughout the document. It is recommended that the authors thoroughly examine and verify the work before submitting it again.

*Thanks for the suggestion. We examined and verified the sentences to avoid the grammar (eg., lines 304, revised clean version).*

Lines 315 – 317: While the authors used "resulted in" in the first statement, the authors used "leads to" in the second sentence. It is recommended that authors stick to the same tense for similar descriptions throughout a single paragraph.

*Thanks for the suggestion. We modified the tense to keep the consistency of descriptions (Line 331, revised clean version).*

Lines 317 – 319: The authors are encouraged to indicate the meaning of the three solid dots shown in the bottom-left corner of Figure 3d. Do they indicate the initial times corresponding to TD, TS, and ST? If yes, please indicate in the figure caption properly. In addition, the authors are encouraged to indicate the meaning of the shaded areas shown in Figure 3a–d. These comments also apply to Figure 4d.
*Yes, the three solid dots in the figure indicate the initial times corresponding to TD, TS, and ST. We modified the caption to explain the meaning of the three dots. Additionally, we explained the meaning of the shaded areas.*

Lines 324 – 325: When describing the differences and/or variations, the authors are encouraged to use numbers or percentages to quantitatively describe it instead of only using the words like "higher", "relatively longer", "weaker intensity", "similar values", or "more intense". This comment also applies to other places with similar issues (e.g., lines 314, 330, 331, 333, 380—382). However, while there are many, the authors are encouraged to change the more relevant ones instead of all.
*Thanks for the suggestion. We agree that using numbers and percentages will provide a clearer and more accurate representation of the data than merely using qualitative terms like 'higher' or 'more intense.' We revised the mentioned sections to include specific numerical data and percentages to quantitatively describe the variations and differences to enhance clarity and precision.*

Lines 411 – 411: The authors are encouraged to revise "6 m s-1" to "6 m s-1".
*Thanks for the suggestion. We modified the "6 m s-1" to "6 m $s^{-1}$ in the revised version (Line 429, revised clean version).*

Lines 454 – 455: Some of the curves (e.g., WRF-ST) are hard to observe in the figure. The authors are encouraged to make the curves easier to observe and add grid lines in the figure (like Figure 7). This comment also applies to Figures 10 and 12.
*Thanks for the suggestions. We modified the colours and line style of the curves in Figure 8 and Figure 10 and added grid lines in Figure 12 to make the curves easier to observe.*

Lines 534 – 534: The dot between "Cha et al" and "(2011)" was missing. The authors are encouraged to revise it as "Cha et al. (2011)".
*We thank the reviewer for this suggestion and agree to adjust to increasing the preciseness of the citation norms (Line 556, revised clean version).*